# Rapid Design and Analysis of Microtube Pneumatic Actuators Using Line-Segment and Multi-Segment Euler–Bernoulli Beam Models

**DOI:** 10.3390/mi10110780

**Published:** 2019-11-14

**Authors:** Myunggi Ji, Qiang Li, In Ho Cho, Jaeyoun Kim

**Affiliations:** 1Department of Electrical and Computer Engineering, Iowa State University, Ames, IA 50011, USA; mji90@iastate.edu (M.J.); qli14@iastate.edu (Q.L.); 2Department of Civil, Construction, and Environmental Engineering, Iowa State University, Ames, IA 50011, USA; icho@iastate.edu

**Keywords:** microtube pneumatic actuator, Euler–Bernoulli beam model, line-segment model, poly(dimethylsiloxane) (PDMS), soft robot

## Abstract

Soft material-based pneumatic microtube actuators are attracting intense interest, since their bending motion is potentially useful for the safe manipulation of delicate biological objects. To increase their utility in biomedicine, researchers have begun to apply shape-engineering to the microtubes to diversify their bending patterns. However, design and analysis of such microtube actuators are challenging in general, due to their continuum natures and small dimensions. In this paper, we establish two methods for rapid design, analysis, and optimization of such complex, shape-engineered microtube actuators that are based on the line-segment model and the multi-segment Euler–Bernoulli’s beam model, respectively, and are less computation-intensive than the more conventional method based on finite element analysis. To validate the models, we first realized multi-segment microtube actuators physically, then compared their experimentally observed motions against those obtained from the models. We obtained good agreements between the three sets of results with their maximum bending-angle errors falling within ±11%. In terms of computational efficiency, our models decreased the simulation time significantly, down to a few seconds, in contrast with the finite element analysis that sometimes can take hours. The models reported in this paper exhibit great potential for rapid and facile design and optimization of shape-engineered soft actuators.

## 1. Introduction

The past decade has seen the rapid rise of soft material-based robots as a new paradigm in robotics [1,2,3,4,5,6,7]. Such “soft robots” are deemed especially useful for biomedicine [8,9] due to their inherent softness, deformability [10,11], and resulting ability to safely manipulate delicate, fragile objects. Of special interest are those based on pneumatic actuation [12,13,14,15,16,17], since other soft robots utilizing heat [18], electromagnetic fields [19], or light [20,21,22,23] for their actuation often entail safety issues associated with high-level voltages [5], electric fields [4], or ultra-violet (UV) light [22]. Regarding the pneumatic soft actuator, the most recent research emphasis has been on its miniaturization for intra-body applications [3]. Balloon-finned micro-fingers [12], the nanofiber-reinforced pneumatic actuator [14], and the magnetically assisted bilayer bending actuator [15] are good examples. However, such small, soft, and safe (S^3^) actuators are difficult to design, optimize, and fabricate, due to their continuum natures and small dimensions.

Recently, we developed a highly unconventional technique for fabricating soft material-based tubes at millimetric and sub-millimetric scales, greatly facilitating the realization of microtube-type S^3^ pneumatic bending actuators [24]. As shown in Figure 1a–c, their bending motion arises from the difference in the microtube’s top and bottom wall thicknesses and the resulting mismatch in their elongation levels under pneumatic inflation. This means that the microtube’s bending pattern can be diversified beyond the simple, circular motion by engineering the microtube’s wall thickness profile as a function of position. Indeed, we amplified the original simple, single-turn bending motion into a complex multi-turn spiraling by placing a hump in the thickness profile [24]. It is expected that more sophisticated shape-engineering could modulate the wall thickness profile multiple times along the microtube’s length, and would bring more variations to the microtube actuator’s bending pattern, greatly increasing its utility.

Currently, the techniques for design and analysis of such multi-segment shape-engineered microtube actuators are very limited, as recently reviewed by Hadi Sadati et al. [25]. The finite element method (FEM) is a powerful scheme for simulating a microtube actuator’s deformation [26,27], but it is very computation-intensive. Euler–Bernoulli’s beam theory has been adopted as a lighter alternative to analyze actuators efficiently [24,28,29,30,31,32,33]. Gorrison et al. utilized it to develop a model that agreed well with the experimentally observed deformation patterns of microtube actuators [28,29]. Shapiro et al. successfully modeled bi-bellow actuators using Euler–Bernoulli models reinforced with material characteristics such as hysteresis [32,33]. They were also used by Shao et al. to model pneumatic bending joints with anisotropic rigidity [31]. To verify the models quantitatively, Shapiro et al. and Hadi Sadati et al. performed comparative studies [25,32]. In our previous work, we adopted this model to optimize a hump’s position [24]. Recently, Wang et al. utilized the line-segment model to predict a pneumatic actuator’s dynamic deformation [34]. The latter two are more suitable for design and analysis of the multi-segment engineered microtube actuator. However, their computational efficiency has not been validated yet.

In this work, we establish a new and simple line-segment model that can effectively deal with multi-segment shape-engineered microtube actuators. Furthermore, we extend our Euler–Bernoulli’s beam theory-based model to enable the analysis of multi-segment microtube actuators. We validate the two models experimentally through quantitative and qualitative comparison studies. The microtube actuators are physically realized and their bending motions are compared under pneumatic actuation against the predictions of the two models. Such a comprehensive, multi-faceted approach for the modeling of soft material-based pneumatic microtube actuators has not been reported yet, to the best of our knowledge. Both the models and the approach adopted here will enrich the field of soft robotics and provide additional tools for the design of soft robots. 

This paper is organized as follows. First, we recapitulate our soft microtube fabrication process. Then, we describe the experimental methods for characterizing their pneumatic actuation and response time. We demonstrate the multi-segment Euler–Bernoulli’s beam model and the line-segment model and verify their predictions against the experimental results quantitatively. A conclusion on the important factors and implications of the work follows.

## 2. Materials and Methods

### 2.1. Microtube Actuator Fabrication

For the experimental validation of the two models, we fabricated highly non-uniform microtube pneumatic actuators using the procedures shown in Figure 2a–e. As the soft material, we utilized poly(dimethylsiloxane) (PDMS, Dow Corning Sylgard 184 Silicon Elastomer, Dow Corning, Midland, MI, USA) with its Young’s modulus (*E*) ~1.4 MPa [11]. The technique exploits the voluntary formation of a PDMS tube around a cylindrical template, followed by gravity-assisted asymmetrization of the tube’s wall thickness distribution and optical fiber jacket remover-enabled “peeling” of the cured PDMS tube. To vary the level of wall thickness asymmetry along the axial direction, we also induced a “beads-on-string” instability, which is frequently observed in liquid-phase (LP)-PDMS [35]. The instability is not suitable for repeatable fabrication of identical microtubes, but is very effective for realizing a wide variety of thickness distributions. More details can be found in our previous reports [36,37].

We first prepared a thin layer of LP-PDMS (~3,500 cP in viscosity) and pre-cured it to increase the viscosity (Figure 2a) [38]. Then, we immersed a cylindrical template in the LP-PDMS (Figure 2b) and lifted it to form a tubular PDMS coating around it (Figure 2c). For this step, commercially available polymer wires with diameters of 250 and 470 μm were utilized as the templates. To induce the wall thickness asymmetry, we intentionally tilted and rolled the LP-PDMS coating under in situ thermal curing at 120 °C, inducing the fluidic instability to the LP-PDMS (Figure 2d). Upon completing the curing process, we obtained a multi-segment, shape-varied microtube, which we peeled off from the template with an optical fiber jacket remover.

For this work, we made eight samples in total. Their total lengths $L_{tot}$ and the top and bottom wall thicknesses $t_{t}$ and $t_{b}$, respectively, are given in Table 1. In general, the microtubes were modeled as cascades of one to four segments, each with its own top and bottom wall thicknesses $t_{ti}$ and $t_{bi}$, respectively, where i is the segment index. Each microtube was attached to a micropipette for pneumatic actuation. As shown in Figure 2f, we could observe a variety of bending patterns by applying pneumatic pressure to the microtube actuator.

### 2.2. Pneumatic Actuation and Characterization

To characterize the microtube actuator’s pneumatic deformation, we used the setup shown in Figure 3. We connected the microtube actuator directly to a pneumatic pump (Pico-Pump, WPI-PV830, World Precision Instruments, Sarasota, FL, USA) to accurately control the inflation. We recorded the resulting bending motion using a camera (Leica DFC-420, Leica Microsystems, Wetzlar, Germany) integrated with an optical microscope (Leica Z16-APO). The structural characteristics of the deformation were extracted through image analysis. The applied pneumatic pressure ranged from 0 to 0.21 MPa, which was sufficient to induce noticeable deformations in the pneumatic microtube actuators. In addition, its motions were recorded through the optical microscope (Appendix A). Their response time was observed to be in the range of 0.18–0.45 s. Microtube actuators with helical tube-wall thickness variations exhibited torsional motions. Since our current work targets modeling of the microtube actuator’s 2D, planar bending motion, we excluded such samples through visual inspection.

### 2.3. Multi-Segment Euler–Bernoulli’s Beam Model

The analytical model based on the Euler–Bernoulli’s beam theory can capture the deformation of the microtube actuator, which contains asymmetries in its wall thickness distribution [28,29,30]. Previously, we applied it to the shape engineering of the microtube actuator to amplify its simple bending motion into a complex spiraling motion [24]. Here, we extend the model further to analyze the bending pattern of the multi-segment shape-engineered microtube actuator in which the wall thickness profile changes multiple times along the length. To that end, the following assumptions were made. First, the bending moment and second moment of area are constant along each section. This can be justified by the short length of each section and the subtleness of the changes in the thickness profile. Second, the Young’s modulus of material is constant along the structure [31,32,33]. 

In the original theory, the deformed coordinates *x* and *y* of the asymmetric cross-section microtube actuator and deviation angle (φ) (Figure 4) are given as
(1)$$\varphi_{i}\left(s\right)=\int_{0}^{s}\frac{M_{i}}{E\cdot I_{i}}ds^{\prime }=\frac{M_{i}\cdot s}{E\cdot I_{i}}$$
(2)$$x\left(s\right)=\int_{0}^{s}\cos\left(\varphi_{i}\left(s^{\prime }\right)\right)ds^{\prime }$$
(3)$$y\left(s\right)=\int_{0}^{s}\sin\left(\varphi_{i}\left(s^{\prime }\right)\right)ds^{\prime }$$
where s is the original length of the pneumatic actuator from 0 to $L_{tot}$, $M_{i}\;\left(i=0,\;1,\;2,\ldots \right)$ is the bending moment of each section, E is the Young’s modulus of the material, and $I_{i}\;\left(i=0,\;1,\;2,\ldots \right)$ is the second moment of area. The bending moment can be calculated with
(4)$$M_{i}=\pi \cdot r_{in}^{2}\cdot d_{ei}\cdot p$$
where $r_{in}$ is the template radius, p is the pneumatic pressure, and $d_{ei}$ is the distance between the neutral axis and the void center (Figure 1b). The neutral axis is the particular longitudinal axis that is neither compressed nor extended, and the pneumatic pressure (p) is applied to the void center. The difference between the operating point of pneumatic pressure and the neutral axis yields the bending moment. Therefore, the increment of thickness difference between the top and bottom tube walls leads to an increase in the distance between the neutral axis and void center ($d_{ei}$), which is in direct proportion to $M_{i}$. Consequently, this triggers the increase of the bending moment that, in turn, enhances the microtube actuator’s deformation.

For this work, we fabricated a variety of axially varying actuators with up to four segments (Table 1). In such situations, the microtube’s thickness distribution parameters $M_{i}$ and $I_{i}$ also become discretized in each segment, as depicted in Figure 5. Accordingly, the integrations for the deformed coordinates become
(5)$$u\left(0<s<a\cdot L_{tot}\right)=\int_{0}^{s}f\left(\frac{M_{0}s^{\prime }}{E\cdot I_{0}}\right)ds^{\prime }$$
(6)$$u\left(a\cdot L_{tot}<s<\left(a+b\right)\cdot L_{tot}\right)=\int_{0}^{s}f\left(\frac{M_{1}s^{\prime }}{E\cdot I_{1}}+\frac{aL_{tot}}{E}\left(\frac{M_{0}}{I_{0}}-\frac{M_{1}}{I_{1}}\right)\right)ds^{\prime }$$
(7)$$u\left(\left(a+b\right)\cdot L_{tot}<s<\left(a+b+c\right)\cdot L_{tot}\right)=\int_{0}^{s}f\left(\frac{M_{2}s^{\prime }}{E\cdot I_{2}}+\frac{bL_{tot}}{E}\left(\frac{M_{1}}{I_{1}}-\frac{M_{2}}{I_{2}}\right)+\frac{aL_{tot}}{E}\left(\frac{M_{0}}{I_{0}}-\frac{M_{2}}{I_{2}}\right)\right)ds^{\prime }$$
(8)$$u\left(\left(a+b+c\right)\cdot L_{tot}<s<L_{tot}\right)=\int_{0}^{s}f\left(\frac{M_{3}s^{\prime }}{E\cdot I_{3}}+\frac{cL_{tot}}{E}\left(\frac{M_{2}}{I_{2}}-\frac{M_{3}}{I_{3}}\right)+\frac{bL_{tot}}{E}\left(\frac{M_{1}}{I_{1}}-\frac{M_{3}}{I_{3}}\right)+\frac{aL_{tot}}{E}\left(\frac{M_{0}}{I_{0}}-\frac{M_{3}}{I_{3}}\right)\right)ds^{\prime }$$
where $I_{i}\;\left(i=0,\;1,\;2,\;\ldots \right)$ is the second moment of area and a,b,c∈R (0, 1) is the fractional section length constrained by a+b+c<1. The function f is a cosine when u=x, and a sine when u=y. 

In this model, we further extend the existing Euler–Bernoulli model to analyze the multi-segment microtube and discretize the bending moment and second moment of area in each segment [24]. The deviation angle of the microtube actuator is related to the ratios between the bending moment ($M_{i}$), Young’s moduli (E), and the second moment of area ($I_{i}$) based on the Euler–Bernoulli’s beam theory [30]. We first calculated the deviation angle of the microtube actuator by performing the integration in Equation (1) along its length, assuming the bending moment and second moment of area to be constant across each segment. Then, we calculated the deformed *x*−*y* coordinate of each segment using Equations (5)–(8) in accordance with the step-wise distributions of $M_{i}$ and $I_{i}$ (Figure 5). We utilized the trapezoidal rules to carry out the integrations over a full cycle for each section’s deformed coordinate while varying the pneumatic pressure.

### 2.4. Line-Segment Model

For line-segment modeling of the microtube actuator, we first divided it into *N* segments as shown in Figure 6a. Applying pneumatic pressure elongates each segment, and the asymmetry in the top and bottom wall thicknesses induce tilting of each segment from the previous one (Figure 6b). Therefore, the bending motion of the microtube actuator can be modeled as successive applications of elongation and tilting. For the former, we simply multiply the elongation factor. 

To implement the latter, we need to find the tilt angle $\theta_{i}$ of each segment which can be utilized in the 2D rotation matrix:(9)$$\left\{\begin{array}{c} x_{i+1}\\ y_{i+1} \end{array}\right\}=\left[\begin{array}{cc} \cos\theta_{i} & -\sin\theta_{i}\\ \sin\theta_{i} & \cos\theta_{i} \end{array}\right]\left\{\begin{array}{c} x_{i}\\ y_{i} \end{array}\right\}$$

To find the elongation factor, we took the material property of PDMS into consideration. A PDMS layer exhibits different elongation rates depending on its thickness. Liu et al. analyzed the thickness-dependent Young’s modulus and stress-strain in PDMS [11]. Young’s modulus can be derived from Hooke’s law (E=σ/ε). The engineering stress σ=F/A, where F is the tensile force and A is the cross-sectional area of the specimen. The engineering strain is defined as $\varepsilon =\left(L-L_{o}\right)/L_{o}$, where $L_{o}$ is the original length of the specimen and L is the elongated length. Gorrison et al. reported that a constant Young’s modulus led to a good agreement with the microtube actuator’s experimental results for low pressure levels (<0.2 MPa) which overlaps with our operation range without consideration of PDMS’s hyperelasticity [28]. Based on the experimental data by Liu et al. [11], we set the range for our ε between 0.5 and 1.2. Our microtube actuators exhibited gravity-induced asymmetries in the top and bottom wall thicknesses. Thus, we utilized different ε’s to find the top and bottom walls’ elongated lengths. The difference between the top and bottom walls’ strain ranges affected the final lengths of the top and bottom walls, which, in turn, induced the actuator’s bending motion. The elongation length equations at the top and bottom are
(10)$$L_{top,i}=L_{o,i}\cdot \varepsilon_{top,i}\cdot \frac{t_{ti}}{t_{bi}}$$
(11)$$L_{bot,i}=L_{o,i}\cdot \varepsilon_{bot,i}\cdot \frac{t_{ti}}{t_{bi}}$$
where $L_{o,i}$ is the initial section length (i = 0, 1, 2, …), $\varepsilon_{top,i}$ and $\varepsilon_{bot,i}$ are the top and bottom strain (depending on the wall thickness), and $t_{ti}$ and $t_{bi}$ are the top and bottom wall thicknesses, respectively, as defined in Figure 1b schematically. 

We formulated the tilt angle $\theta_{i}$ as a function of all these characteristics. Note that the eccentricity of the void center $d_{ei}$, defined schematically in Figure 1b, played an important role in the formulation. The final tilt angle equation is
(12)$$\theta_{i}=\tan^{-1}\left(\frac{L_{bot,i}-L_{top,i}}{r_{in,i}}\right)\cdot \frac{p_{o}}{p_{max}}\cdot \left(1+d_{ei}\right)$$
where $p_{o}$ and $p_{max}$ are the initial and maximum pressures that we applied to the pneumatic actuator. Obviously, the tilt angle decreases as the pneumatic pressure applied to the structure becomes lower. By combining these, we established a simple line-segment model capable of making quick predictions on the shape-dependent bending pattern.

## 3. Results and Discussion 

To validate the Euler–Bernoulli’s beam model and line-segment model for the multi-segment shaped-engineered microtube actuator, we carried out comparison studies. To that end, we selected three microtube actuators (numbers 1, 2, and 3) as our representative samples. They were all multi-segmented with different numbers of segments, as specified in Table 1. They also differed in their thickness distributions and overall lengths. Their maximum bending motions were achieved at 0.14, 0.14, and 0.17 MPa, respectively. Additionally, the three samples’ minimum response time constants required for maximum bending motion were measured to be 0.18, 0.45, and 0.38 s, respectively. 

Using such a variety of microtube actuators, we aimed to show that the modeling results agreed well with those from the experiments regardless of the segment counts or thickness distributions. The results for the Samples 1, 2, and 3 are shown in Figure 7, Figure 8 and Figure 9, respectively. The 1st, 2nd, and 3rd rows demonstrate the results from the experiment, the multi-segment Euler–Bernoulli model, and the line-segment model, respectively. Overall, the results from the two models exhibited good qualitative agreements with the experimental results.

For a more quantitative comparison, we utilized the maximum bending angle, defined as the angle between the horizontal line and the tangent of the beam at the end point at the final stage of inflation, as the criterion. This is an important figure-of-merit that determines the maximal bending level of the microtube bending actuator. The results are displayed in Table 2. The experiment results showed that the maximum bending angles of Samples 1, 2, and 3 were 306.4°, 175.7°, and 273.2°, respectively. The maximum bending angles of Samples 1, 2, and 3 that were extracted from the results of the multi-segment Euler–Bernoulli’s beam model were 332.7°, 180.8°, and 243.8°, respectively. For the maximum bending angles of Samples 1, 2, and 3 from the line-segment model results, we obtained 292.7°, 198.2°, and 283.1°, respectively. For Samples 1 and 2, the deviations from the experimental results were within ±10%. For Sample #3, it was still within ±11%, confirming that the modeling results agreed well with the experimental results.

These semi-analytical and numerical models, based on multi-segment Euler–Bernoulli beam theory and line-segments, respectively, can be used as rapid methods for the design and analysis of multi-segment, highly shape-engineered microtube actuators. In terms of computational efficiency, the two methods can significantly shorten the simulation time, possibly down to a few seconds, in comparison with the FEM-based methods typically take hours to calculate, even for uniform microtube actuators [26,27]. The multi-segment Euler–Bernoulli and line-segment models also share a potential to enhance their accuracies through an increase in the number of segments. In that respect, the multi-segment model exhibits a relative superiority over the Euler–Bernoulli model, since the latter requires a re-formulation for each addition of a new section. The line-segment model will also be more suitable for analyzing torsional, 3D bending motions in the future. 

## 4. Conclusions

In conclusion, we have developed a simple line-segment model as well as a multi-segment version of Euler–Bernoulli’s beam theory-based model to design and optimize the complex bending motion of non-uniform, highly shape-engineered microtube pneumatic actuators. For the former, the key enabling factor was the proper setup of the “tilt function”. We established our own, which was based on experimental observations and basic soft matter mechanics. For the latter, we attempted a multi-segment formulation of the existing Euler–Bernoulli model. For its validation, we fabricated multiple non-uniform microtube actuators and recorded their bending patterns. The predictions from the line-segment and multi-segment Euler–Bernoulli models and the experimental results agreed well on the final bending pattern. In terms of the maximum bending angle, which is a quantitative figure of merit, the error was within 11%. Given the minute dimensions of the microtube actuators (8–12 cm in length and 0.25–0.5 mm in diameter) and subtle changes in the thickness distribution, such a high level of agreement is very meaningful. Further, the two models exhibited a high potential to significantly shorten computational time down to a few seconds in contrast with the more computation-intensive FEM-based methods. These two new models are capable of expediting the design and optimization of axially non-uniform microtube actuators with complex bending patterns, widening the scope of their future applications as S^3^ actuators.

## Figures and Tables

**Figure 1 micromachines-10-00780-f001:**
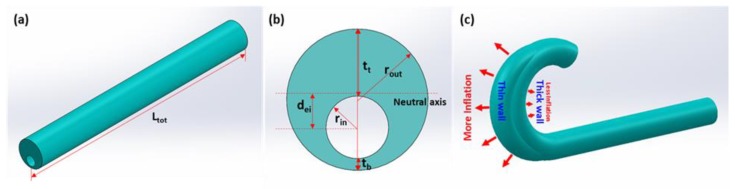
(**a**) A schematic diagram of a microtube actuator. (**b**) Definitions of the microtube actuator’s structural parameters in a cross-sectional view. (**c**) A schematic diagram showing the origin of the microtube actuator’s bending motion upon pneumatic inflation.

**Figure 2 micromachines-10-00780-f002:**
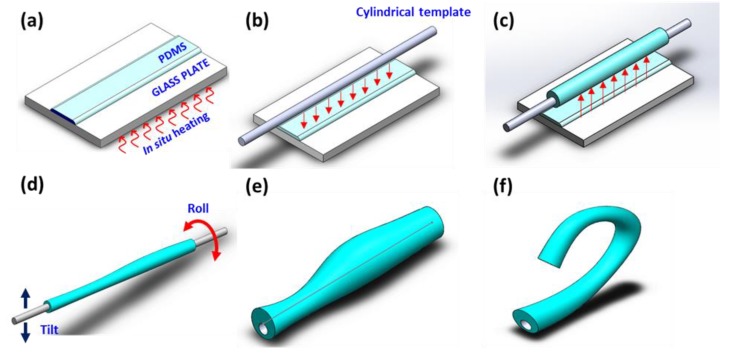
(**a**) In situ pre-curing of liquid-phase poly(dimethylsiloxane) (LP-PDMS) on a glass plate. (**b**) Preparation of a cylindrical template and its immersion into the pre-cured LP-PDMS. (**c**) Lift-up of the PDMS-coated template. Waiting time is needed for gravity-assisted asymmetrization. (**d**) Rolling and tilting of the template to induce the fluidic instability and further asymmetrize the microtube. (**e**) Peel-off of the completed microtube (drawn not to scale). (**f**) One end of the microtube is sealed to make the microtube inflatable. Application of pneumatic pressure deforms it to induce bending motions in various patterns.

**Figure 3 micromachines-10-00780-f003:**
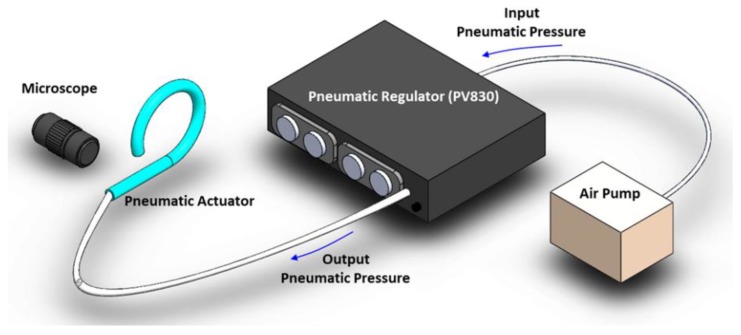
Experimental setup for pneumatic actuation and its characterization.

**Figure 4 micromachines-10-00780-f004:**
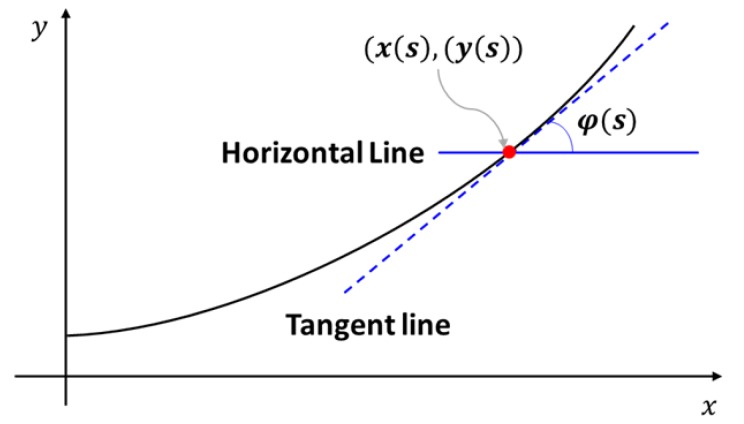
A schematic diagram of the deformed x−y coordinates and the deviation angle φ(s).

**Figure 5 micromachines-10-00780-f005:**
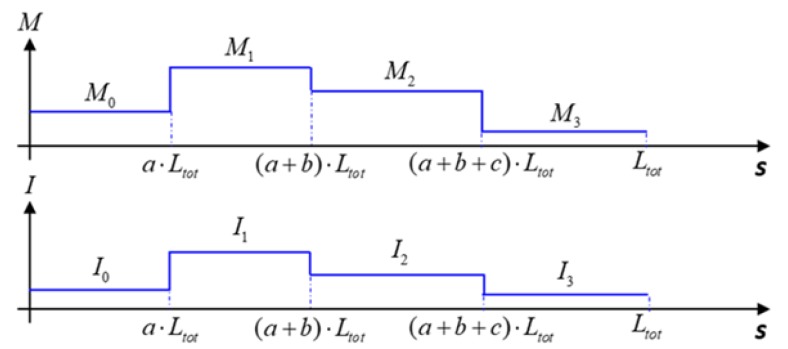
Step-wise distributions of the non-uniform microtube actuator’s bending moment ($M_{i}$) and second moment of area ($I_{i}$ ).

**Figure 6 micromachines-10-00780-f006:**
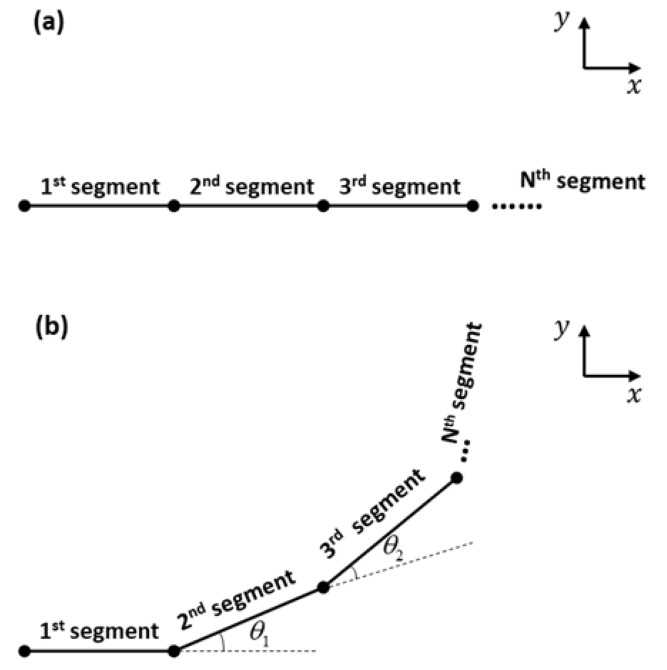
Line-segment method configuration. (**a**) Initial state of the actuator. (**b**) Pneumatic pressure applied state of the actuator with bending angle $\theta_{i}$.

**Figure 7 micromachines-10-00780-f007:**
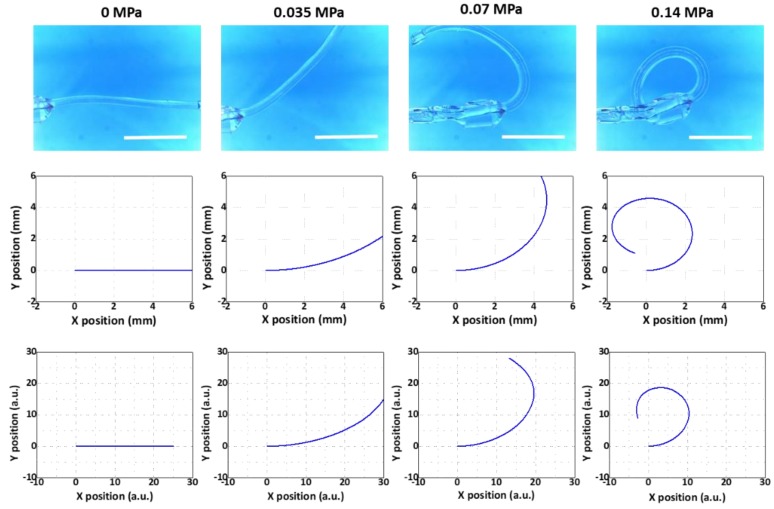
Comparison between the experimental model (**top**), the multi-segment Euler–Bernoulli’s beam model (**middle**), and the line-segment model (**bottom**) (Sample 1 in Table 1). Scale bars: 5 mm.

**Figure 8 micromachines-10-00780-f008:**
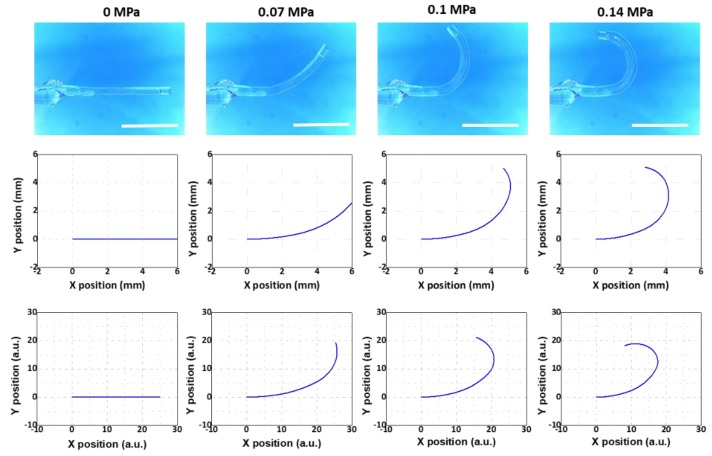
Comparison between the experimental model (**top**), the multi-segment Euler–Bernoulli’s beam model (**middle**), and the line-segment model (**bottom**) (Sample 2 in Table 1). Scale bars: 5 mm.

**Figure 9 micromachines-10-00780-f009:**
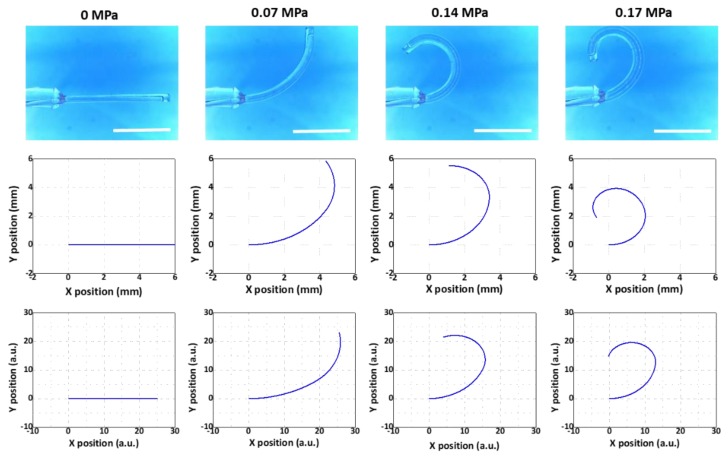
Comparison between the experimental model (**top**), the multi-segment Euler–Bernoulli’s beam model (**middle**), and the line-segment model (**bottom**) (Sample 3 in Table 1). Scale bars: 5 mm.

**Table 1 micromachines-10-00780-t001:** Microtube actuator specification.

Sample			#1	#2	#3	#4	#5	#6	#7	#8
Template Diameter (μm)			470						250	
$L_{tot}$ (mm)			12.29	8.38	8.78	12.06	9.17	9.57	8.88	8.80
Fractional Section Length ($L_{i}∕L_{tot}$)		0	0.43	0.33	0.61	0.17	0.60	0.17	0.75	1.00
		1	0.24	0.36	0.39	0.41	0.30	0.83	0.13	-
		2	0.23	0.31	-	0.42	0.10	-	0.12	-
		3	0.10	-	-	-	-	-	-	-
Wall Thickness (μm)	0	$t_{t0}$	134	200	129	164	126	101	92	63
		$t_{b0}$	67	140	72	72	100	38	51	42
	1	$t_{t1}$	166	113	113	250	142	167	123	-
		$t_{b1}$	72	67	62	72	59	38	82	-
	2	$t_{t2}$	211	185	-	123	80	-	82	-
		$t_{b2}$	67	62	-	72	67	-	51	-
	3	$t_{t3}$	139	-	-	-	-	-	-	-
		$t_{b3}$	67	-	-	-	-	-	-	-

**Table 2 micromachines-10-00780-t002:** The maximum bending angles from the experimental and modeling results.

Sample	#1	#2	#3
Experiment	306.4°	175.7°	273.2°
Multi-segment Euler–Bernoulli’s beam model	332.7°	180.8°	243.8°
Line-segment model	292.7°	198.2°	283.1°

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
