# Peer review of "Rapid Design and Analysis of Microtube Pneumatic Actuators Using Line-Segment and Multi-Segment Euler–Bernoulli Beam Models"

_micromachines, 2019, doi:10.3390/mi10110780_

Round 1

Reviewer 1 Report

This paper presents a soft pneumatic micro-tube actuator to generate the bending motion. This has been undertaken through rapid design analysis and multi-segment E-B beam models. Both simulation and experimental results are presented and compared. This paper is well written with sufficient results. This reviewer suggests the authors to accommodate the following comments before making a final decision.

Material properties used in this work need to be provided (for example, LP-PDMS, etc) Please use SI standard units for all variables. Please present the response time taken from the initial position to the final position. This is very important criterion in applications. How to achieve the plane motion? Is there no torsional motion? This should be clearly addressed. The presentation of the experimental animation as a supplementary material will be very useful to understand the actuating motion of the proposed soft structure. What is the main difference of this work from the related previous works or the authors’ previous works? In other words, what is the technical novelty of this work? Actuator Principle, New Material Actuator, Bending Motion Generation, Etc? Conclusion needs to be rewritten in details by presenting quantitative values achieved in this work.

Reviewer 2 Report

The authors discuss the problem of design and analysis of microtube pneumatic actuators. The topic is for interest for the journal readers. The reference list is proper and up to date. The paper is well structured. The language is proper, but a proofread is recommended. Some terms are used inappropriate (for example formula is not an equation, like it is used in equation 12). The novelty of work is not clear. The main idea of microtubes is already published in the paper "Microrobotic tentacles with spiral bending capability based on shape-engineered elastomeric microtubes". The modelling part uses well-known equations. Please highlight the added value. The paper is hard to follow due to the lack of symbol descriptions. I suggest to introduce a list of symbols. Please add all the used values in order to help the reader to obtain you results.

Reviewer 3 Report

Introduction: State of the art is adequate, most relevant articles on soft robots, micro-pneumatic actuation and their design/analysis methods are properly referenced. As this is not the first time the Euler-bernoulli beam model is used for modeling bending pneumatic actuators, references and short commentary should be provided on relevant publications, indicatively:

[1] Y. Shapiro, et al., "Bi-bellows: Pneumatic bending actuator", Sensors and Actuators A: Physical, 2011.
[2] Y. Shapiro, et al., "Modeling a Hyperflexible Planar Bending Actuator as an Inextensible Euler–Bernoulli Beam for Use in Flexible Robots", 2015.
[3] T. Shao, et al., "Basic Characteristics of a New Flexible Pneumatic Bending Joint", Chinese Journal of Mechanical Engineering, 2014.
[4] S. M. Hadi Satati, et al., "Mechanics of Continuum Manipulators, a Comparative Study of Five Methods with Experiments", TAROS 2017.

The lack of design and analysis of microtube actuators is highlighted but contribution and novel points should also be compared to the above articles.
Minor: A paragraph on the structure of the article is always appreciated as the last part of the intro.

Section 2: Fabrication method is clearly presented and the gravity-induced wall asymmetry is properly described.

Repeatability should be commented, how easy it is to fabricate a pneumatic actuator of the same characteristics and especially the wall thickness asymmetric properties? As this is the most important aspect for real-life fabrication and testing of such actuators, commentary should be provided on the repeatability aspect.

While the discretization of the Bernoulli-Euler model for each segment is highlighted, any further differentiation from the standard model should be better highlighted in this Section.

Minor: a short commentary should be provided on the safety of the assumption that the second moment of area and the bending moment is constant across each segment.

"we calculated the strain epsilon varies from 0.5 to 1.2.." minor grammar/syntax mistakes were identified, please correct.

How does the epsilon range 0.5-1.2 affect your calculations considering the incorporation of two bottom/top strains? Could you please clarify this statement?

More information on the tilt angle formulation from (10) and (11) to (12) should be added to increase readability. t_{ti} and t_{bi} should be defined in the text.

Section 3: The results adequately show the validity of the presented models. The results summarized in Table 2 show a differentiation between tested cases, while no clear 'winner' is presented. Do the authors believe that both models share the same potential? More emphasis on the comparative results should be given.

No comparison is been made on any computational aspects between the models, while no comparison is performed between the proposed approach and finite element approaches seen in related literature. These data would significantly increase the quality of the contribution.

Conclusion: Please incorporate more comments regarding the novel points and acquired results, as this seems more of a repetition of the Abstract.

Round 2

Reviewer 1 Report

This paper is now acceptable.

Author Response

The authors thank the reviewer for the valuable comments.

Reviewer 2 Report

The authors did not revised the manuscript in accord with the recommendations. The main issue, the novelty, is not solved. This section must be included in the introduction section and the whole work must prove this novelty statement. Regarding the inappropriate use of some terms: you have to use "equation" instead of "formula".
